# Transmission Removal from a Single OSN-Shared Glass Mixture Image

**Heng Yao *, Zhen Li and Chuan Qin**

School of Optical-Electrical and Computer Engineering, University of Shanghai for Science and Technology, Shanghai 200093, China; lz1029655099@163.com (Z.L.); qin@usst.edu.cn (C.Q.)

* Correspondence: hyao@usst.edu.cn; Tel.: +86-136-6148-8750

**Abstract:** Photographs taken through glass often reflect the photographer or the surroundings, which is very helpful in uncovering information about the photograph. Various lossy operations performed on images over online social networks (OSNs), such as compression and resampling, pose a great challenge for transmission layer removal. This paper proposes a self-attention-based architecture for image enhancement over OSNs, to ensure that the downloaded glass mixture image can show more information about the reflection layer than the original image. Transmission layer removal is then achieved using a two-stage generative adversarial network. We also add attention to the transmission layer in the mixture image and use the gradient and color block information in the next stage to extract the reflection layer. This method yielded a gain of 0.46 dB in PSNR, 0.016 in SSIM, and 0.057 in LPIPS, resulting in an effective improvement in the visual quality of the final extracted reflection layer images.

**Keywords:** transmission removal; vision transformer; OSN-shared image; glass mixture image

## 1. Introduction

When taking a photograph through glass, we can obtain a glass mixture image that contains a transmission layer and a reflection layer. The operation of recovering the reflection layer from the mixture image is defined as the transmission removal problem. Figure 1 illustrates a schematic diagram of the glass mixture image separation problem, where transmission removal (also referred to as reflection separation or reflection extraction) focuses more on exploring features that are not easily detectable in the blended image. Studies based on transmission layer removal can be effective in aiding the identification of photo capture scenarios (e.g., photographer and location information). In traditional methods based on statistical analysis, exploring the differences between the two layers is a prerequisite for achieving the removal of one of the layers.

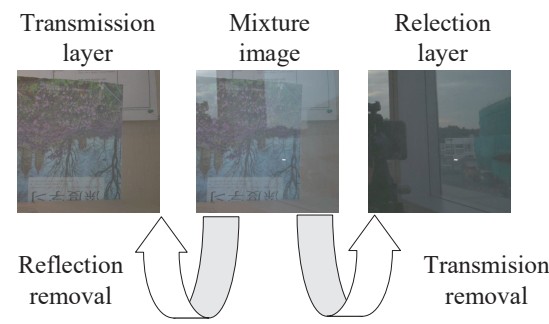

**Figure 1.** Schematic diagram of glass mixture image separation problem.

Today, everyone can share daily images over online social network (OSN) platforms, and according to statistics, the amount of image data shared over Instagram is about

1.3 billion images per day (How many pictures are there (2023): Statistics, trends, and forecasts. Available: https://photutorial.com/photos-statistics/, accessed on 19 November 2023), in addition to some of the other OSNs, such as WhatsApp, Facebook, Snapchat, and WeChat. Therefore, studying social media shared images can help understand the practical significance of interpersonal activities. When an image is uploaded to a platform, the platform will accordingly perform a series of processing processes on the image [1–3]. These operations can be roughly summarized as follows: (1) resize the image: images uploaded by users may have various sizes, and one way to process an image is to resize it to make it more suitable for displaying on social media; (2) enhance image quality: uploaded images may appear blurry or unclear due to low light or other reasons, and OSNs can enhance the quality of images through manipulating them, such as increasing contrast, adjusting brightness and color saturation; (3) crop the image: uploaded images may contain unnecessary information, such as backgrounds or other clutter. By cropping an image, the subject can be highlighted and made more appealing; (4) add filters: filters are often used on OSNs to make images more attractive, and they can also change the color and contrast of images and add special effects, such as blurring or black and white effects; (5) recognize faces: social media can use face recognition techniques to identify faces in images and manipulate them; (6) protect user privacy: social media can protect user privacy by processing images. For example, social media can use blurring techniques to hide sensitive information in images, such as addresses, license plate numbers, or faces. This means that the re-downloaded image data are not exactly the same as the original image, in terms of pixels and structure, which greatly increases the difficulty of understanding the information contained in the image. Therefore, it is a challenge to effectively remove the transmission layer when a glass mixture image is uploaded to OSNs.

This paper considers a scenario in which people share images containing glass reflections over OSNs. Figure 2 shows a visual comparison of OSN shared images, where from the point of view of visual quality, it is difficult to determine the difference between the two images (a) and (b), which indicates that the overall distribution does not change during the network transfer, so we analyze the differences in the frequency domain. We perform DCT on these two images separately, and according to (c) and (d), the red box indicates the low-frequency information of the image, while outside the red box mainly contains the high-frequency information of the image, which is selected and marked with a green box for the convenience of analysis. In contrast, before and after the OSN transfer, the red box does not have a significant impact on the low-frequency information content of the image; however, during the acquisition of the high-frequency information in the green box, a part of the visibility will be lost, which means that when further analyzing the non-dominant reflection layer information, it is difficult to use the relevant information as a feature to output the reflection layer directly. Therefore, the manipulation of the image by the social media platform has caused significant changes to the original image after sharing.

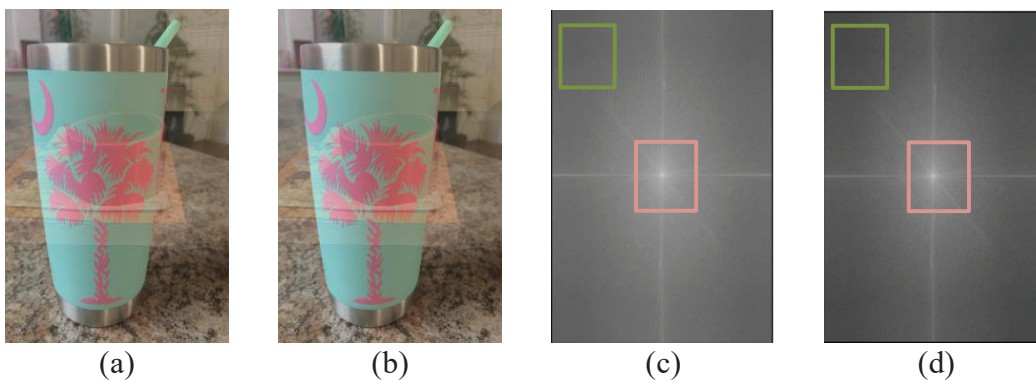

    (a)                (b)                (c)                (d)

**Figure 2.** Visual comparison of glass mixture images shared over OSNs. (**a**) original image; (**b**) OSN shared image; (**c**) spectrogram of (**a**) after discrete cosine transform (DCT), and (**d**) spectrogram of (**b**) after DCT.

To further verify the effect on the glass mixture image, we compared the results of the two images directly using an existing reflection layer extraction experiment, and the comparison results are shown in Figure 3. In general, the information of the OSN-shared image was incomplete compared to the results of the original input. For the original mixture image, the non-dominant reflection layer image becomes sparse as the information in each part of the image becomes tighter, due to the overlaying between the different image layers after the OSN platform propagation process, and we hope to solve the layer separation problem adaptively for an OSN-shared glass mixture image by building a network model methodology.

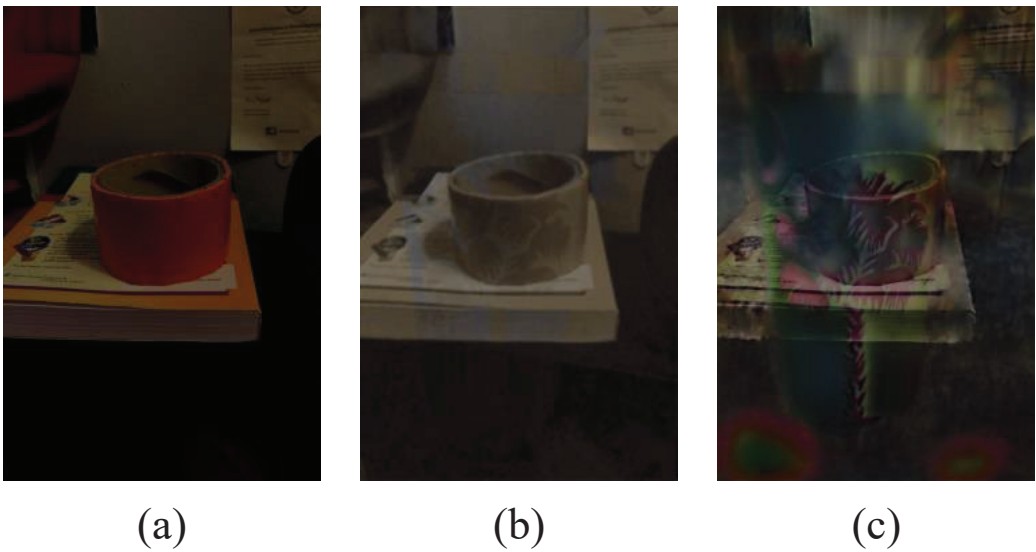

(a)  (b)  (c)

**Figure 3.** An example of reflection layer extraction. (**a**) ground-truth reflection layer image; (**b**,**c**) predicted reflection layer from the original and the OSN-shared glass mixture images, respectively.

Incorporating the motivations mentioned above, in this paper, we intend to improve the understanding of the original image after the OSN sharing using a network with an attention mechanism, and meanwhile, to regenerate the reflection layer image using an adversarial generation network architecture with enhanced feature correlation. The contributions of our work can be summarized as follows:

- We consider the impact of social networks on the current transmission layer removal problem in glass mixture images, since the transmission removal problem is essential as an aid to shooting scene and photographer recognition;
- We propose a method for achieving image feature enhancement based on a self-attention mechanism, to ensure that the enhanced glass mixture image further improves the correlation between the reflection layer information;
- We design a two-stage generative adversarial network to remove the transmission layer. To fully use the valuable information in OSN-shared images, we also add attention to the transmission layer information in the mixture image and apply the estimated gradient information and the color patch information to the later stages of the transmission layer removal process;
- The method yielded a gain of 0.46 dB in PSNR, 0.016 in SSIM, and 0.057 in LPIPS, effectively improving the visual quality of the final extracted reflection layer images.

The remainder of this paper is organized as follows: Section 2 introduces the related works about transmission removal and image processing shared over OSN. Sections 3 and 4 describe the proposed method and analyze the experimental results, respectively, and Section 5 concludes this paper.

## 2. Related Works

This study deals with the reflection layer extraction of OSN-shared glass mixture images, which can be fundamentally understood as the enhancement of OSN images in a specific glass reflection scenario. This section presents the transmission layer removal and OSN image processing problems separately.

### 2.1. Transmission Layer Removal

Many efforts have successfully separated this component of the reflection layer from the glass mixture image. More specifically, the extracted reflection layer has been applied to the actual scene; for example, Wan et al. [4] proposed and solved a face-image reflection removal problem. They recovered the important facial structures by incorporating inpainting concepts into a guided reflection removal framework, which took two images as the input and considers various face-specific priors. Li et al. [5] developed a network structure based on a deep encoder–decoder repetition-reduction network (RRNet). The authors found that the redundant information increased the difficulty of predicting images in the network; thus, they used mixed reflection image cascaded edges as input to the network. Chang et al. [6] proposed joint reflection removal and depth estimation from a single image by concatenating intermediate features. Li et al. [7] proposed an iterative boost convolutional long short-term memory (LSTM) network that enabled cascaded prediction for reflection removal. The estimates of the transmission and reflection layers could be iteratively refined by boosting each other's prediction quality, and information across the cascade steps was transferred using an LSTM.

The mixture image reflection extraction process is an extension of the study of glass image de-reflection and is mainly divided into statistical feature algorithms and deep learning algorithms, both of which distinguish the difference in features between the two image layers in the mixture image. Levin and Weiss [8] analyzed the statistical feature distribution of the two image layers to determine the boundary and thus achieve separation. For deep learning algorithms, it is difficult to utilize the statistical information of the images, which requires some additional a priori knowledge as an aid; for example, the gradient information of the two layers can be used as the essential feature, to distinguish the two due to their different priorities in the mixture image. Zhang et al. [9] used gradient information and proposed a gradient loss function as a way to separate the two image layers. Similarly, Wan et al. [10,11] and Chang et al. [6] constructed a predicted gradient network and then used this predicted gradient as a prior to assist in image layer extraction. In addition, the final experimental results were refined by continuous iterations of deep learning. Li et al. [7] proposed a cascade network that allowed two image layers to be used as a prior for each other, thus iteratively highlighting the information in the predicted image. Li et al. [12] recovered reflection layer images that were as close as possible to the ground truth images by analyzing the texture dimension as well as style dimension.

### 2.2. Image Processing over OSNs

There have been no systematic studies on social media sharing of mixture images, and the most relevant to this study are some robust watermarking algorithms used against JPEG compression [13–18]. In addition, a portion of the work involves studying a series of operations specific to transmitting images on OSN platforms. Specifically, Wu et al. [19] used deep convolutional neural networks (DCNNs) to predict the impact of social media network platforms on uploaded images. They designed an image forgery detector and proposed a new training scheme for robust image falsification detection of OSN transmission. Wu et al. [20] identified anomalous regions by constructing MT-Net, to assess the degree of difference between local features and reference features. Mayer and Stamm [21] introduced forensic region similarity, to determine whether two images contain the same or different synthetic traces on block-level regions.

## 3. Proposed Method

This section describes in detail the process of separating the reflection layer from the glass mixture images obtained from social media platforms through computer vision techniques.

### 3.1. Overall Network Architecture

Figure 4 illustrates an overall schematic diagram of the proposed method, where the reconstruction network is a cascade architecture consisting of a mixture image feature-enhancement network **Net$_1$** and a two-stage reflection separation network with transmission removal **Net$_2$**. The specific function models of **Net$_1$** and **Net$_2$** are as follows:

$$\begin{aligned}
\mathbf{Net_1} &: (\mathbf{I'}, \mathbf{E_B}) = G_1(\mathbf{I}), \\
\mathbf{Net_2} &: (\hat{\mathbf{B}}_\mathbf{C}, \hat{\mathbf{B}}_\mathbf{E}) = G_{2\text{-}1}(\mathbf{I'}, \mathbf{E_B}), \\
\hat{f}(\mathbf{R}) &= G_{2\text{-}2}(\mathbf{I}, \hat{\mathbf{B}}_\mathbf{C}, \hat{\mathbf{B}}_\mathbf{E}),
\end{aligned} \tag{1}$$

where **I** is the shared input image, **I'** is the enhanced image, **E$_B$** is the enhanced edge image, $G_1$ is the feature enhanced network, $G_{2\text{-}1}$ and $G_{2\text{-}2}$ are the two-stage extraction networks, $\hat{\mathbf{B}}_\mathbf{C}$ and $\hat{\mathbf{B}}_\mathbf{E}$ are both predicted transmission features, where the former and latter represent the color distribution and texture distribution of the predicted image, respectively. $\hat{f}(\mathbf{R})$ is the predicted reflection feature.

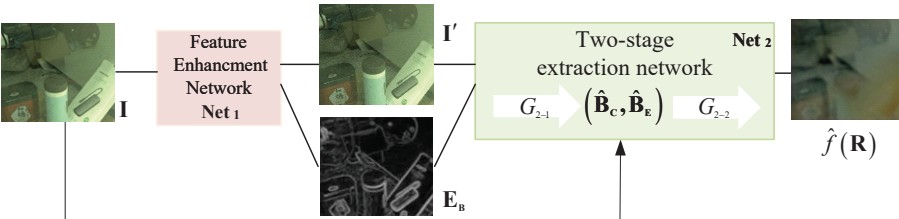

**Figure 4.** Overall schematic diagram of the proposed method.

(1) **Net$_1$**

The attention mechanism focuses on enhancing small-scale features using the global attention mechanism, while extracting gradient profile information about the transmission layer as additional information. We use the Transformer architecture [22] as the global attention mechanism, which has yet to be developed for visual processing applications, with the most significant limitation being the consistent chunking of images when processed as input [23]. This imposes strict requirements on the size of the input. Thus, to meet the requirements of this study, we only use **Net$_1$** for the network of enhanced image features. Secondly, although the mixture image has been greatly changed, the transmission layer in the mixture image still dominates. Hence, the feasibility exists to perform processing directly using means related to edge extraction. At the same time, according to some existing studies, using gradient edge extraction, the underlying information can satisfy the requirements for the gradient edge, and if too much deep-level information is used, this will instead cause information redundancy when regenerating. Therefore, we only refer to the underlying convolutional features in **Net$_1$** for the edge information, and the specific analysis will be described in the ablation experiment.

(2) **Net$_2$**

The features obtained in the previous section are developed for further learning, with the ultimate goal of exploring effective ways to achieve the removal of the transmissive layer using both the enhanced features and the gradient profile. Specifically, the paired information obtained in the previous part of the network is closely related to the transmittance layer. Therefore, in order to better consider the visual effect of the predicted reflection layer, we do not use a single step for predicting the reflection layer, but first, perform a prediction operation for the transmission layer, and then use this predicted transmission

layer information containing edge and color block information as an auxiliary prior for guiding the reflection layer separation.

Sections 3.2 and 3.3 introduce **Net₁** and **Net₂** in detail, respectively.

### 3.2. Transformer-Based Image Feature Enhancement Network

(1) Network architecture

Compared with the traditional CNN architecture, a transformer architecture can represent the global features obtained from the shallow layer based on the attention mechanism. However, the standard transformer calculates the global self-attention between all tokens, which results in a very large amount of computation, especially for high-resolution images. At the same time, a transformer is weak at capturing local context information. In this way, a locally-enhanced window (LeWin) is introduced into the overall U-shaped structure [23]. In this paper, we pay more attention to the estimation associated with the original image, as well as the edges in terms of features, so we adopt a different architecture for these two parts.

Figure 5 illustrates the feature enhancement operation for the input image. It can be seen that we use two different decoders as the output architecture of the augmented network, and at the same time, these two parts are different in their specific implementation process. The network is mainly reflected in the application of features and the constraints on the generated results. For the enhanced image features, it is not realistic to impose too many constraints on the spatial structure, because the enhanced image itself is applied to the information of the input image. On the other hand, since the global structure of the image is weakened after the input image has been processed by the OSN platform, the loss of structure is of less significance.

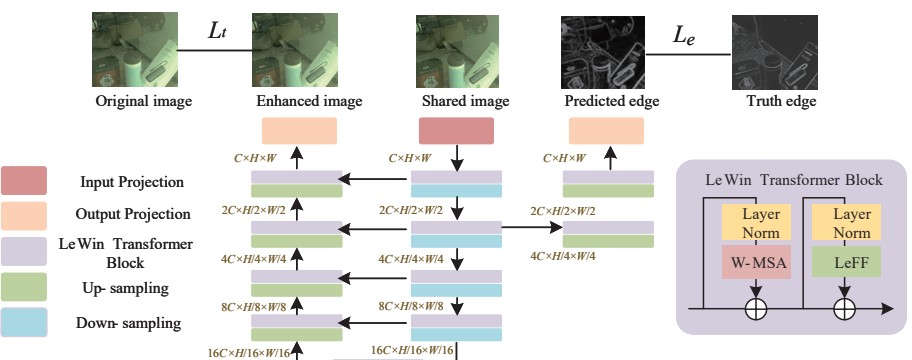

**Figure 5.** Feature-level image enhancement network.

Figure 6 illustrates in detail the architecture of the modules included in the enhancement network. As shown in Figure 6a, window multi-head self attention (W-MSA) [24] is different from the global self-attention of the standard transformer. We perform self-attention in a non-overlapping local window, which can effectively reduce the amount of calculation. Specifically, for a two-dimensional input feature $X(C \times H \times W)$, we first split $X$ into non-overlapping blocks of size $M \times M$, represented as $X = \{X^1, X^2, X^3, \ldots, X^N\}$, where $N = HW/M^2$ is the number of blocks. Next, self-attention is performed on all of these window features. The $k$-th header calculation process can be described as $Y_k^i = \text{Attention}\left(X^i W_k^Q, X^i W_k^K, X^i W_k^V\right)$, where $i = 1, 2, \ldots, N$. Finally, this is combined to obtain the final result $\hat{X}_k = Y_k^1, Y_k^2, \ldots, Y_k^N$.

Considering the limitation of the feedforward network in the transformer for capturing local contextual information and the importance of neighboring pixels to the overall restoration of the image and partial contour information, depth convolution is added to the feedforward network to capture local context information. The structure is shown in Figure 6b. In the feedforward network, a layer of convolution operation is added to capture the local context information of the input features. Specifically, each token is first

processed with fully-connected convolution to increase the number of channels, and then the token is re-converted into a 2D feature map using a $3 \times 3$ convolution to capture local information. The 2D feature map is then re-pulled into a 1D token, and the number of channels is re-decreased using fully-connected convolution, to make this consistent with the number of channels before input.

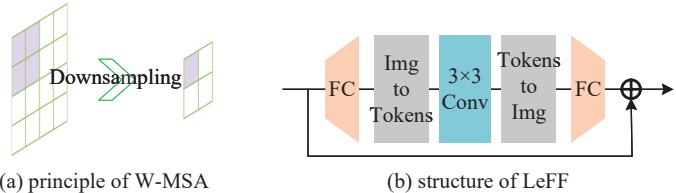

(a) principle of W-MSA  (b) structure of LeFF

**Figure 6.** Two components of LeWin transformer block.

(2) Loss function

The regenerated image $\hat{f}(\mathbf{R})$ is used as the input for the next stage of the network, which needs to be made closer to the ground-truth scene image in the regeneration process. Thus, through the above analysis, we use $L_1$, perceptual loss $L_P$ [25], and total variance loss $L_{TV}$ to place further constraints on the texture and finally form the following loss function:

$$L_t = \omega_1 L_1 + \omega_2 L_P + \omega_3 L_{TV}, \tag{2}$$

where $\omega_1$, $\omega_2$, and $\omega_3$ are parameters for weighing partial losses, which are set as 5, 3, and 6, respectively.

Among them, we introduce the total variance loss $L_{TV}$, which enhances the details of the image and reduces the effect of noise by penalizing the differences between adjacent pixels in the image. The total variance $J$ is the integral of the gradient magnitude, which can be expressed as

$$J(\mathbf{u}) = \iint_{x,y \in \text{ image}} |\nabla \mathbf{u}| \mathrm{d}x\mathrm{d}y = \iint_{x,y \in \text{ image}} \sqrt{\mathbf{u}_x^2 + \mathbf{u}_y^2} \mathrm{d}x\mathrm{d}y \tag{3}$$

where $x$, $y$ represent the horizontal and vertical directions of the image, respectively; and $\mathbf{u}_x$, $\mathbf{u}_y$ are the gradients of the pixels in the $x$, $y$ directions, respectively. Since the total variance of the image contaminated by noise is larger than that of the uncontaminated one, eliminating the noise means making the total variance smaller. At the same time, for the image, the integral of the continuous domain is equivalent to the summation of the discrete domain, so it is transformed into the following loss function $L_{TV}$:

$$L_{TV} = \sum_{i,j} \sqrt{\left(\hat{f}(\mathbf{R})_{i,j} - \hat{f}(\mathbf{R})_{i+1,j}\right)^2 + \left(\hat{f}(\mathbf{R})_{i,j} - \hat{f}(\mathbf{R})_{i,j+1}\right)^2}, \tag{4}$$

where $i$ and $j$ represent the pixel coordinates in the image, and the sharpness and details of the image are enhanced through calculating and minimizing the differences between pixels and their neighbors.

The second output of the augmented network **Net$_1$** is the gradient profile information of the transmission layer. Since the gradients are coherent on the whole and the final output is a one-dimensional image, few outliers interfere with the fit. Thus, to reflect the coherence of the edge information, the mean squared error loss $L_{MSE}$ is used here for the constraint. Since the gradient information is more concentrated in this region, using $L_{MSE}$ is not influenced by the interference information and can achieve a better fitting process at the gradient level. The expression is as follows:

$$L_e = \omega_4 L_{MSE}, \tag{5}$$

where $\omega_4$ was set as 5 in our experiments.

### 3.3. Two-Stage Reflection Extraction Network

(1) Network architecture

In the last section, we progressively regenerated the enhanced features of the glass mixture image after the platform processing and simultaneously output the associated gradient profile information as the input for the next step in implementing the network. Since the transformer performs image vision operations by downscaling the input tensor, this results in the unfolded tensor occupying different spaces depending on the input size; in other words, it is easy to for the system capacity to be insufficient when the image size is too large.

Therefore, in response to the above analysis, we instead use the adversarial generative network as our main framework to implement the subsequent operations concerning transmission removal. We reconstruct the two-stage network to extend more prior knowledge as guidance. Figure 7 illustrates the process of transmission layer removal after using the acquired enhanced features. We initially use the residual network for the prediction of the transmission layer, and then link the three features together again to form the feature map input for the second-stage generation network.

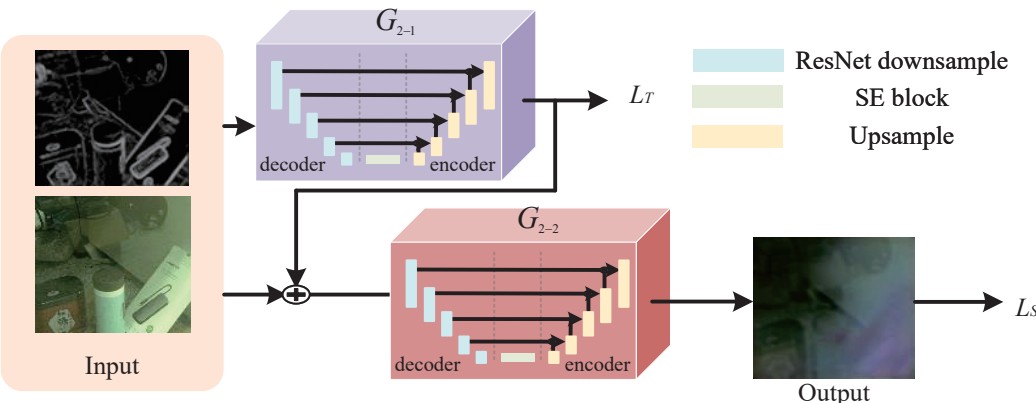

**Figure 7.** Two-stage transmission layer removal model.

(2) Loss function

For the output results, we construct a discriminator network to discriminate the authenticity of the final results. Based on this, we determine the loss of the first-stage predicted transmission layer $L_T$ and the loss of the second-stage predicted reflection layer $L_S$, respectively.

$L_T$ is defined as

$$L_T = \lambda_1 L_P + \lambda_2 L_1, \tag{6}$$

where $\lambda_1$ and $\lambda_2$ are the parameters, which are set as 5 and 2, respectively.

In addition, for the content of the image reflection layer, we use texture loss $L_{\text{texture}}$ [26] instead of pixel-level loss and weaken the correlation with the transmission layer using the exclusion loss $L_{\text{excl}}$ [9]. The second-stage predicted reflection layer loss is constructed as

$$L_S = \lambda_{11} L_{\text{excl}} + \lambda_{21} L_{\text{texture}} + \lambda_{31} L_{\text{TV}}, \tag{7}$$

where $\lambda_{11}$, $\lambda_{21}$, and $\lambda_{31}$ are the parameters set as 2, 2, and 5, respectively.

Ultimately, the specific loss function $L_B$ is composed of $L_T$ and $L_S$ together as

$$L_B = L_T + L_S. \tag{8}$$

### 3.4. Dataset

Many benchmark training datasets exist for research on glass mixture images, and these training sets help to provide a uniform qualitative measure when validating a method. However, these existing data are incomplete when faced with more complex scenarios.



Therefore, we needed to perform additional data augmentation in the case of some shared factors on social media platforms involved in this experiment. First, we uploaded the mixture images to a mainstream social media network (taking Facebook as an example). We downloaded the images again and divided the data for training and validation. Finally, we obtained a training dataset of 1951 quadratic data, a valid dataset of 491 quadratic data, and a test dataset of 50 quadratic real scenes. Note that the above original datasets containing triple images were from [27].

We found that during the actual downloading process, the names of the original files as well as the arrangement order changed due to the limitations of the platform. We used information matching to reintegrate these images and rename them, so as to ensure that the one-to-one correspondence could be maintained in the experiments.

Figure 8 shows five sets of illustrations from our reconstructed dataset and compares the pixel sizes of these five sets of images before and after passing through the social media network, where the first row represents the original input and the second row represents the shared image, keeping the same size between the corresponding objects. It can be seen that there was a significant decrease in pixel occupancy for the shared images, and since pixels are the product of their size and resolution, in the case of a constant size, the shared images are resolution-decreasing.

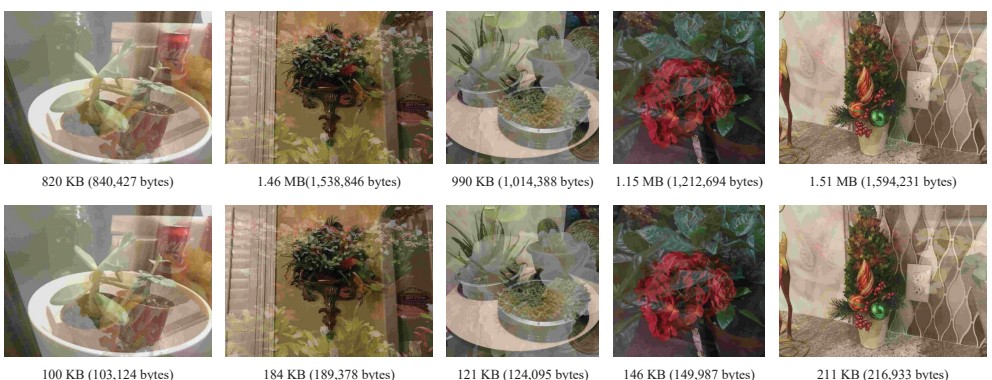

| 820 KB (840,427 bytes) | 1.46 MB(1,538,846 bytes) | 990 KB (1,014,388 bytes) | 1.15 MB (1,212,694 bytes) | 1.51 MB (1,594,231 bytes) |
| 100 KB (103,124 bytes) | 184 KB (189,378 bytes) | 121 KB (124,095 bytes) | 146 KB (149,987 bytes) | 211 KB (216,933 bytes) |

**Figure 8.** Example of the dataset.

## 4. Experimental Results

### 4.1. Implementation and Training Details

We implemented our model using Pytorch. To satisfy the input window size of the attention mechanism, in the training phase, all source and target sizes were clipped randomly to $128 \times 128$. We trained for 50 epochs with a batch size of 16. The separation model learned from random initialization using the RMSProp optimizer [22] with a learning rate of $5 \times 10^{-5}$. The whole training process converged in approximately 14 h using a single GPU GeForce GTX 3090 Ti for the 1951 image pairs from the training data. We implemented random cropping for images with more than 640,000 pixels. In the test network, a cyclic calculation of batch size 1 was used to calculate the mean value.

### 4.2. Comparison of Results

In this section, we compare the final generated results with related reflection extraction experiments. The main image measures we use are the peak signal-to-noise ratio (PSNR), structural similarity (SSIM), and visual perceptual similarity (LPIPS). In order to evaluate the performance of the scheme unbiasedly, we compare it with some cutting-edge studies on reflection layer separation.

Figure 9 presents a comparison of the final generated results, where (a) represents the input image; (b) represents the real scene image of the reflection layer; (c–f) are the images of Li et al. [28], Chang et al. [29], Yang et al. [30], and Zhang et al. [9]'s results; and (g) is our results. It can be seen that our experiments were able to achieve the elimination effect as much as possible for the issue of transmission removal from glass mixture images

compared to the other comparison experiments, and it can be seen from the comparison experiments that the transmission layer was still difficult remove in the final generated results. The reason for this was that the differences between the two image layers need to be better understood. In order to use the average results obtained from 50 sets of comparison images, we used three evaluation image quality metrics to measure the final generated results in the comparison experiments; the results are listed in Table 1.

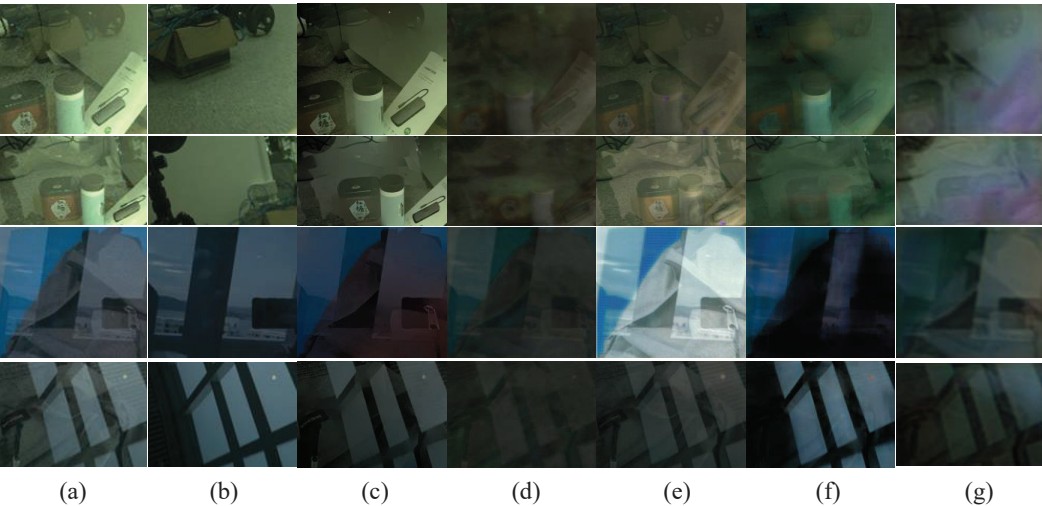

| (a) | (b) | (c) | (d) | (e) | (f) | (g) |

**Figure 9.** Comparison of the final transmission removal results. (**a**) represents the input image; (**b**) represents the real scene image of the reflection layer; (**c–f**) are the images of Li et al. [28], Chang et al. [29], Yang et al. [30], and Zhang et al. [9]'s results; and (**g**) is our results.

**Table 1.** Statistical comparison of quality indicators (best results are bolded).

|  | LPIPS ↓ | SSIM ↑ | PSNR (dB) ↑ |
| --- | --- | --- | --- |
| Li et al. [28] | 0.663 | 0.517 | 17.820 |
| Chang et al. [29] | 0.735 | 0.566 | 19.837 |
| Yang et al. [30] | 0.772 | 0.479 | 10.122 |
| Zhang et al. [9] | 0.674 | 0.718 | 21.883 |
| Ours | **0.606** | **0.734** | **22.343** |

Note that we adopted the recently proposed learned perceptual metric (LPIPS) [31] as an error metric. This measures perceptual image similarity using a pretrained deep network. This metric learns the reverse mapping of generated images to ground truth and forces the generator to reconstruct the reverse mapping of authentic images from fake images and to prioritize the perceived similarity between them. LPIPS is more consistent with human perception than the traditional methods (such as L2/PSNR, SSIM, and FSIM). The lower the LPIPS value, the more similar the two images; otherwise, the greater the difference. Without loss of generality, we also chose SSIM and PSNR as metrics to compare references. As can be observed in the table, our results showed gains of 0.46 dB in PSNR, 0.016 in SSIM, and 0.057 in LPIPS compared to the related studies, and an improvement was achieved in all three metrics.

Figure 10 shows an LPIPS value line chart for 50 test image sets, and although there are some image groups with higher values, our results were generally more stable and relatively low.

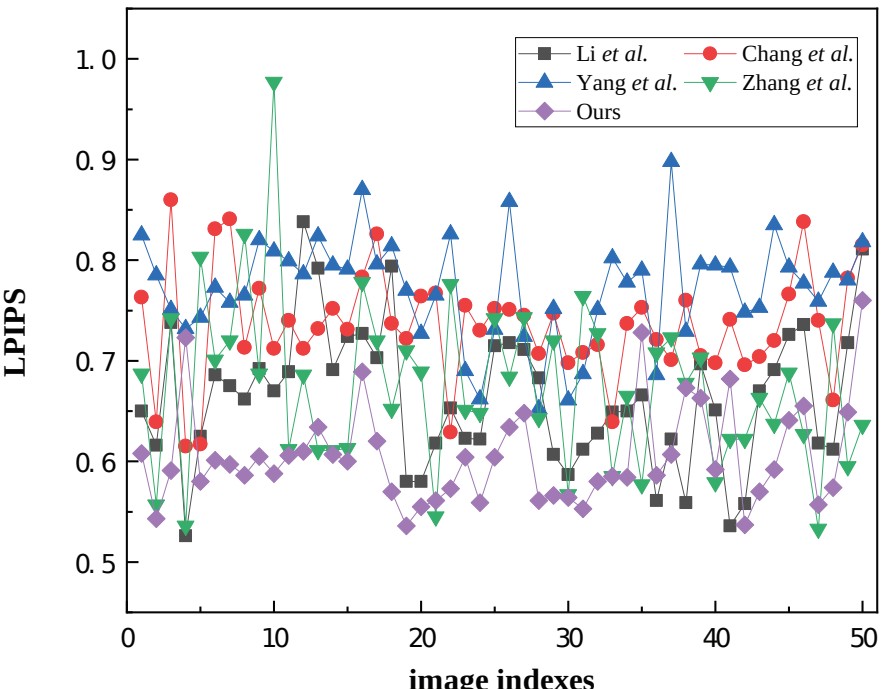

**Figure 10.** LPIPS value line chart of 50 test image sets. The comparison methods involved include Li et al. [28], Chang et al. [29], Yang et al. [30], Zhang et al. [9], and ours.

### 4.3. Ablation Study

While constructing the network, we noted that the image output at each node could also be used as a basis for optimizing the effectiveness of the network, i.e., to achieve a better visual quality for these images. Therefore, we utilized the image evaluation metrics to measure the strength of our final network.

Figure 11 demonstrates the effect of extracting gradient information in our feature enhancement stage. In order to better serve as input features for the next stage, we focused on the integrity and continuity of the edge information, so we set different depth networks for learning, where (d) only used the higher-level information network architecture with richer gradient information. As can be seen, compared to (b) and (c), the edge image (d) was sufficiently clear and exhibited a better visual output when only the underlying information was used.

Figure 12 shows the ablation comparison experiments in the final acquired reflection layer, where the single-stage network in (c) and (d) refers to the network that did not indirectly predict the transmission layer information but directly predicted the reflection layer, and (h) represents the final completed experimental configuration. Through comparison, it can be seen that the complete loss condition was more capable of removing the information in the transmission layer, thus preserving and highlighting the content of the reflection layer. For a more quantitative analysis, Table 2 lists the corresponding quality indicator values. It can be seen that (h) possessed a clearer visual quality than the other experimental results, which demonstrates that the proposed method could obtain more information about the reflection layer after removing the transmission layer.

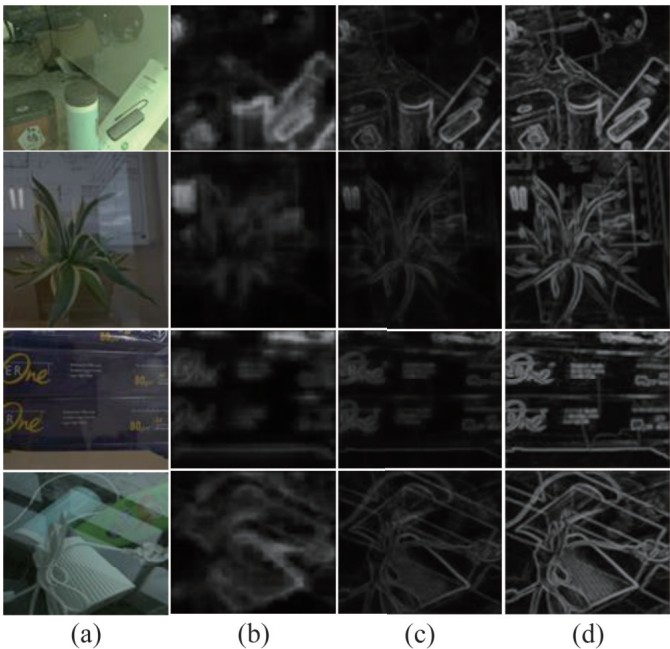

**Figure 11.** Ablation experiment for gradient profile prediction; (**a**) represents the input image and (**b**–**d**) represent the edge gradient ablation results by setting different network depths; to be specific, the size of (**b**) was $16C \times H/16 \times W/16$, (**c**) was $8C \times H/8 \times W/8$, and (**d**) was $4C \times H/4 \times W/4$.

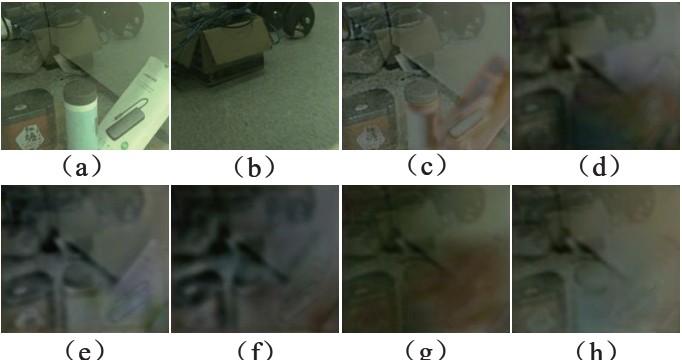

**Figure 12.** Ablation experiment for reflection layer extraction; (**a**) represents the input image, (**b**) represents the ground-truth background image of the reflection layer, (**c**,**d**) represent a single-stage network without $L_{\text{texture}}$ and $L_{\text{TV}}$, respectively; (**e**–**g**) represent a two-stage network without $L_{\text{excl}}$, $L_{\text{texture}}$, and $L_{\text{TV}}$, respectively; and (**h**) represents the final completed experimental configuration.

**Table 2.** Quantitative analysis of quality indicators for the images in Figure 12.

|  | LPIPS ↓ | SSIM ↑ | PSNR (dB) ↑ |
|---|---|---|---|
| Figure 12c | 0.578 | 0.710 | 20.080 |
| Figure 12d | 0.679 | 0.674 | 17.423 |
| Figure 12e | 0.677 | 0.696 | 18.198 |
| Figure 12f | 0.747 | 0.699 | 18.616 |
| Figure 12g | 0.676 | 0.693 | 18.010 |
| Figure 12h | 0.626 | 0.743 | 22.444 |

Figure 13 shows the predicted images for the high-resolution images, where the high-resolution images are the input images that were not transmitted by the OSN, and the predicted images were the result of retraining the network architecture, which can be seen to have removed the transmission layer to a greater extent in terms of visual recovery. For

the purpose of this study, this means that there is still room for improvement in terms of image super-resolution and recovery of auxiliary edge information.

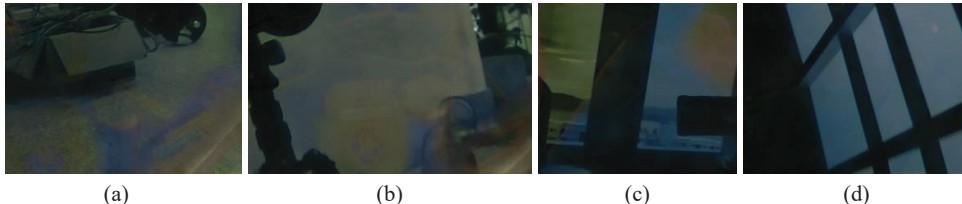

| (a) | (b) | (c) | (d) |

**Figure 13.** Ablation experiment for transmission removal from original high-resolution image; (**a**–**d**) represent the predicted images corresponding to Figure 9.

## 5. Conclusions

In this paper, we analyzed the impact of social media sharing on reflection layer separation studies for glass mixture images and proposed a targeted image processing tool based on enhanced image features, using the enhanced features as learning objects to achieve the final goal of transmission layer removal. Specifically, we used the transformer architecture, with its attention mechanism feature, to enhance the input image features, while using the underlying features to output the subsequent a priori gradient profile information as an ancillary product in the enhancement process. Immediately after, based on further learning of these two parts using a two-stage generative network, we finally achieved transmission layer removal for the OSN-shared glass mixture image. The experimental results demonstrated the efficacy of the proposed network, and the rationality of the network setup was also validated by outputting the results of the steps and measuring the metrics through an ablation study.

**Author Contributions:** Conceptualization, H.Y. and Z.L.; methodology, H.Y. and Z.L.; software, Z.L.; validation, H.Y. and Z.L.; formal analysis, H.Y. and Z.L.; investigation, H.Y. and C.Q.; resources, Z.L.; data curation, Z.L.; writing—original draft preparation, Z.L.; writing—review and editing, H.Y. and C.Q.; visualization, Z.L.; supervision, C.Q.; project administration, H.Y. and C.Q.; funding acquisition, H.Y. and C.Q. All authors have read and agreed to the published version of the manuscript.

**Funding:** This research was funded by the Natural Science Foundation of China grant number 62172281, 62172280, U20B2051, 61702332, the Natural Science Foundation of Shanghai grant number 21ZR1444600, and the STCSM Capability Construction Project for Shanghai Municipal Universities grant number 20060502300. The APC was funded by the Natural Science Foundation of China grant number 62172281 and STCSM Capability Construction Project for Shanghai Municipal Universities grant number 20060502300.

**Institutional Review Board Statement:** Not applicable.

**Informed Consent Statement:** Not applicable.

**Data Availability Statement:** Data are contained within the article.

**Acknowledgments:** The authors would like to thank the anonymous reviewers for their valuable suggestions, which helped to improve this paper.

**Conflicts of Interest:** The authors declare no conflict of interest.

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
