# Peer review of "Transmission Removal from a Single OSN-Shared Glass Mixture Image"

_applsci, doi:10.3390/app132312779_

Round 1
Reviewer 1 Report
Comments and Suggestions for Authors
The paper is well organized. The research problem is significant and inherently difficult. With such poor results, however, I would suggest trying to solve an easier problem: transmission removal from a high-resolution image. At the very least, the paper should include results for this easier problem.
Additional comments:
1. Figure 10 should be changed to a box plot or results of statistical significance tests because it is difficult to see.
2. More details about the dataset need to be provided, such as the source of the images and the content of the images.
3. The numbers of images in the dataset given in section 3.4 and 4.1 are different. Please check.
4. In section 4.1, some detail is missing from "All source and target sizes are then clipped randomly to 128."
Comments on the Quality of English LanguageMinor language editing is needed.
Reviewer 2 Report
Comments and Suggestions for Authors
General Comments:
The paper deals with transmission layer removal in glass-mixture images and presents a deep-learning approach based on self-attention mechanism for image enhancement over online social networks. Good improvements in terms of PSNR and SSIM have been obtained versus existing methods while extracting reflection-layer images from glass-mixture images. The topic is important in image processing and the proposed approach is useful.
Specific Comments:
1. Please clarify how the network in Figure 6b can capture local context information.
2. It is unclear how edge information is kept intact by the proposed attention mechanism.
3. Please explain how the variance loss function in Equation (3) can differentiate between noise and correlative relations among pixels. Is there a threshold for outliers? Noise performance would be helpful if added to Section 4.
Round 2
Reviewer 2 Report
Comments and Suggestions for Authors
The Authors have addressed the Reviewer’s comments carefully, especially as related to noise processing and keeping edge information.
The current version of the paper is useful and suitable for publication.